# Order-Independence Without Fine Tuning

**Reid McIlroy-Young**[*]
Department of Computer Science
Harvard University

**Katrina Brown**
Department of Computer Science
Harvard University

**Conlan Olson**
Department of Computer Science
Columbia University

**Linjun Zhang**
Department of Statistics
Rutgers University

**Cynthia Dwork**
Department of Computer Science
Harvard University

## Abstract

The development of generative language models that can create long and coherent textual outputs via autoregression has lead to a proliferation of uses and a corresponding sweep of analyses as researches work to determine the limitations of this new paradigm. Unlike humans, these '*Large Language Models*' (LLMs) are highly sensitive to small changes in their inputs, leading to unwanted inconsistency in their behavior. One problematic inconsistency when LLMs are used to answer multiple-choice questions or analyze multiple inputs is *order dependency*: the output of an LLM can (and often does) change significantly when sub-sequences are swapped, despite both orderings being semantically identical. In this paper we present **Set-Based Prompting**, a technique that *guarantees* the output of an LLM will not have order dependence on a specified set of sub-sequences. We show that this method *provably* eliminates order dependency, and that it can be applied to *any* transformer-based LLM to enable text generation that is unaffected by re-orderings. Delving into the implications of our method, we show that, despite our inputs being out of distribution, the impact on expected accuracy is small, where the expectation is over the order of uniformly chosen shuffling of the candidate responses, and usually significantly less in practice. Thus, Set-Based Prompting can be used as a '*dropped-in*' method on fully trained models. Finally, we discuss how our method's success suggests that other strong guarantees can be obtained on LLM performance via modifying the input representations.

Code is available at `github.com/reidmcy/set-based-prompting`.

## 1 Introduction

Recent advances in machine learning have led to a paradigm shift, as new training methods, which rely on massive scale datasets to create general purpose 'base models', outperform specialized single purpose models (Achiam et al., 2023; Radford et al., 2019). A particularly notable development is in natural language processing, where self-supervised learning has led to a cornucopia of transformer based models that can string together multiple next

---

[*]Corresponding author `reidmcy@seas.harvard.edu`

38th Conference on Neural Information Processing Systems (NeurIPS 2024).

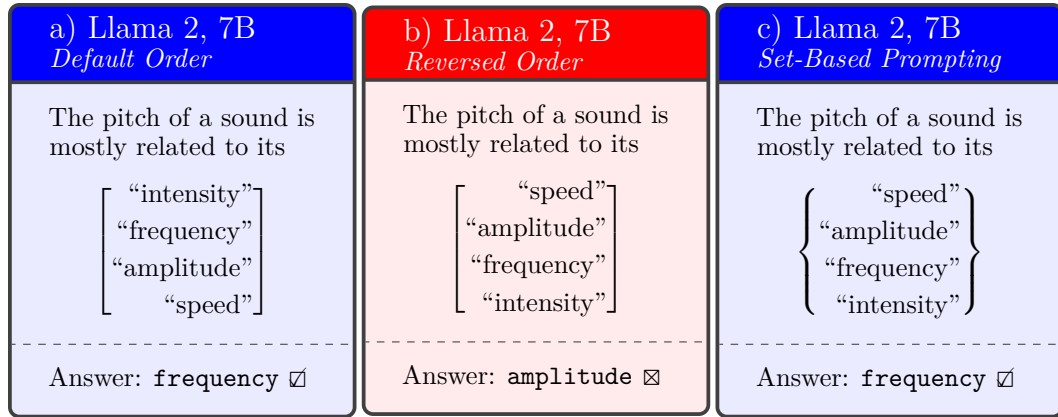

Figure 1: Illustration of order dependency in Llama 2, 7B. Using the order provided by (Measuring Massive Multitask Language Understanding) (MMLU) (Hendrycks et al., 2020) Llama 2 gets the question correct as seen in a), but if the order is reversed for the questions Llama 2, 7B predictions the wrong answer. In c) we use Set-Based Prompting to remove the ordering of the answers and Llama 2, 7B once again gets the question correct.

token predictions, via autoregression, to generate coherent text that approximates a human response. These generated responses are not human and exhibit non-human behavioral quirks due to the limitations of the transformer based LLM architecture. Some of these, like the inability to reason about letters, are due to the choice of tokenization (the process of converting text into a sequence of numbers that can be used as an input) which breaks words into multiple character chunks (McCoy et al., 2023), while others like the *Lost Middle Phenomenon* are limitations of the training data (N. F. Liu et al., 2024). The solutions to these limitations are known (An et al., 2024), although often impractical (Sutton, 2019).

There are other limitations of LLMs that appear to be fundamental to the design of LLMs. One pernicious issue is the *order dependency problem*. This is the phenomenon where the output of an LLM can be significantly altered by changing the order of inputs, even when the re-ordering should have an identical response; see figure 1 for an example. This *order dependency problem* is well studied for multiple choice questions (Pezeshkpour et al., 2023; Zheng et al., 2024), but can happen on any task.

Having systems that make or aid in decision making that are sensitive to factors of the input that are orthogonal to the goals of the operators presents significant concerns. Consider an LLM that reads medical papers and summarizes them to a doctor (Van Veen et al., 2024). If shuffling the papers changes the summary can the doctor trust the model's responses? Additionally, *order dependency* presents algorithmic fairness concerns, such as in a case where an LLM being used to compare candidates has a slight bias for the first candidate (Adian Liusie, 2024; Li et al., 2023).

In this paper we present a solution to the order dependency problem. Our method **Set-Based Prompting** solves this problem by removing the ordering information from the inputs to the LLM, for a specified set of *sub-sequences*. We refer to this as *prompting* to emphasize that our method does not modify the underlying model, it only changes the input. These sub-sequences of multiple tokens are (from the model's perspective) run in parallel, leading to the output being agnostic as to the ordering of the sub-sequences[2]. We show that our method guarantees the parallel treatment of sub-sequences for arbitrary transformer based LLMs. We then test it on a variety of models on multiple choice question tasks and show that while parallel sub-sequences can impact performance, the impact is usually within that caused by re-ordering the sub-sequences when run without our method. We also discuss how the success of Set-Based Prompting suggests many future potential contributions.

---

[2]We sometimes refer to these as *parallel* sub-sequences to emphasize that order information has been removed, so we have a set of sub-sequences instead of a list.

## 2    Related Works

Multiple choice questions (MCQs) serve as an important task format in the evaluation of large language models (LLMs). Previous work has shown that modern LLMs are vulnerable to re-ordering of options in MCQs. Pezeshkpour et al., 2023 identify inherent 'positional bias' in the MCQ setting, where LLMs prefer to select options in specific positions as answers, and propose reordering the options to position the most likely candidates in specific locations within the sub-sequence. This is viable in the case where both the prior positional bias and the correct answers are known a priori, but not necessarily in more general settings. Zheng et al., 2024 identifies a slightly different problem of 'token bias', wherein LLMs prefer to select specific option IDs A/B/C/D as answers. They propose estimating prior token bias via a label-free, inference-time debiasing method, which separates the model's prior bias for option IDs from the overall prediction distribution. Adian Liusie, 2024 likewise identifies positional bias in the setting of pairwise comparisons of natural language generation, and suggests debiasing using estimates of prior positional bias derived from observing comparison outcomes for a subset of candidates. These methods all assume the existence of a set of comparable samples with which to estimate prior ordering biases. Other work looking at how modifying word choice affects the model's outputs (Salewski et al., 2024) show that this issue extends beyond order-dependency, but this matches how humans answer questions (Tjuatja et al., 2023) so should be less unintuitive for non-experts.

There is also an example of this order-dependence being used to probe the training data of LLMs. Oren et al., 2023 shows that the ordering of questions in the training data affects the ordering during inference, allowing for detection of training data contamination.

One implication of this MCQ order-dependence is that it makes the relative comparisons of the performance of different models on benchmarks such as MMLU less reliable, as different evaluation runs may employ different MCQ orderings within the test sets (Alzahrani et al., 2024).

## 3    Set-Based Prompting

To provably ensure order-independence we introduce **Set-Based Prompting**, an elegant 2-pronged modification to the model at inference time touching, respectively, the attention mask and the positional encoding. As these areas are not usually considered to be variable, this pushes the model slightly out of distribution. Furthermore, we hypothesize that some output quality degradation may occur since in the order-independent case there is both less computation and less information, as no tokens in the set of parallel sequences can attend to tokens in other parallel sequences. However, we hypothesize that base models are robust to minor changes, and output coherence will not be noticeably impaired. In fact, to see the impact of our methods requires many queries. We begin with a review of the attention mechanism's attention mask and the positional encoding system used in transformer-based LLMs. Then we will show how we can use a *non-triangular* mask combined, together with a modified positional encoding — our two prongs — to make the attention block unable to determine the order of sub-sequences. This results in completely removing all the ordering information *between* sub-sequences, making the model's outputs perfectly order invariant.

### 3.1    Background

**Attention Mask.** The attention mechanism is a means of taking in an arbitrary number of inputs and *attending* to them in a weighted manner. This allows for longer sequences to be input and processed into a single output. When processing text, the attention mechanism is used on each input token in turn, creating an embedding for each token separately.

In practice, we say the tokens can see all other tokens in the input, but when generating text the tokens in the input are usually treated as if they are the next to be generated and future tokens are masked output. This is seen in LLMs including GPT-2 (Radford et al., 2019) and Llama (Touvron, Lavril, et al., 2023) with their self-attention mechanism, wherein the outputs for a certain position in a sequence are based only on the known outputs at

Order-Dependant Prompting

| $j =$ | 1 | 2 | 3 | 4 | 5 | 6 | 7 |
|---|---|---|---|---|---|---|---|
| $i = 1$ the | 1 | 0 | 0 | 0 | 0 | 0 | 0 |
| $i = 2$ aptly | 1 | 1 | 0 | 0 | 0 | 0 | 0 |
| $i = 3$ quick | 1 | 1 | 1 | 0 | 0 | 0 | 0 |
| $i = 4$ light | 1 | 1 | 1 | 1 | 0 | 0 | 0 |
| $i = 5$ reddy | 1 | 1 | 1 | 1 | 1 | 0 | 0 |
| $i = 6$ brown | 1 | 1 | 1 | 1 | 1 | 1 | 0 |
| $i = 7$ fox | 1 | 1 | 1 | 1 | 1 | 1 | 1 |
|  | $\boldsymbol{p}(i,1)$ | $\boldsymbol{p}(i,2)$ | $\boldsymbol{p}(i,3)$ | $\boldsymbol{p}(i,4)$ | $\boldsymbol{p}(i,5)$ | $\boldsymbol{p}(i,6)$ | $\boldsymbol{p}(i,7)$ |
|  | + | + | + | + | + | + | + |
| $\boldsymbol{X} =$ [ | the, | aptly, | quick, | light, | reddy, | brown, | fox ] |

Set-Based Prompting

| $j =$ | 1 | 2 | 3 | 2 | 3 | 4 | 5 |
|---|---|---|---|---|---|---|---|
| $i = 1$ the | 1 | 0 | 0 | 0 | 0 | 0 | 0 |
| $i = 2$ aptly | 1 | 1 | 0 | **0** | **0** | **0** | 0 |
| $i = 3$ quick | 1 | 1 | 1 | **0** | **0** | **0** | 0 |
| $i = 2$ light | 1 | **0** | **0** | 1 | 0 | 0 | 0 |
| $i = 3$ reddy | 1 | **0** | **0** | 1 | 1 | 0 | 0 |
| $i = 4$ brown | 1 | **0** | **0** | 1 | 1 | 1 | 0 |
| $i = 5$ fox | 1 | 1 | 1 | 1 | 1 | 1 | 1 |
|  | $\boldsymbol{p}(i,1)$ | $\boldsymbol{p}(i,2)$ | $\boldsymbol{p}(i,3)$ | $\boldsymbol{p}(i,2)$ | $\boldsymbol{p}(i,3)$ | $\boldsymbol{p}(i,4)$ | $\boldsymbol{p}(i,5)$ |
|  | + | + | + | + | + | + | + |
| $\boldsymbol{X}_s =$ [ | the, | {[aptly, | quick ], | [light, | reddy, | brown ]}, | fox ] |
|  | $\boldsymbol{X}_{start}$ | $s_1$ | | $s_2$ | | | $\boldsymbol{X}_{end}$ |

Figure 2: Visualization of the differences between order-dependant prompting (left) and Set-Based Prompting (right). Our input is the prompt '*the aptly quick light reddy brown fox*' and '*aptly quick*' is in parallel to '*light reddy brown*'. Each row represents a query to an attention block (we treat each word as a token), with the index of the query given by $i$. $\boldsymbol{X}$ and $\boldsymbol{X}_s$ give the set of values over which the query is attending. $\boldsymbol{p}(i,j)$ is the vector-valued positional encoding which is added to the word's embedding. The center of the diagram is the attention mask $\boldsymbol{M}_{j,i}$.

previous positions and not on future positions. This is implemented by multiplying the inputs by a mask vector that is 0 for all future tokens. When these mask vectors are stacked (for parallel computation), they form a triangular matrix, the attention mask matrix for that input. Figure 2 shows this attention mask for a sequence of 7 tokens.

**Positional Encoding.** Different language models use different methods of representing the position of input tokens within a sequence. The original transformer introduced by Vaswani et al., 2017 proposed absolute positional embeddings, which involves assigning a specific value vector at each time step, or position, and then adding element-wise those values to the token embedding. This is the positional encoding method used in models such as `GPT-2`.

In contrast, most state of the art LLMs including `lama2/3`, and PaLM (Anil et al., 2023), use rotary positional embeddings (RoPE) as introduced by Su, Lu, et al., 2023, which encode absolute positional information with a rotation matrix and naturally incorporate explicit relative position dependency in the self-attention formulation.

## 3.2 Provable Order-Independence via Set-Based Prompting

We now describe Set-Based Prompting , our technique for modifying the positional encoding to provide parallel (order-free) representation of sequences of text, and prove that it ensures order-independence. Set-Based Prompting can be applied to both absolute and RoPE embeddings, on both `GPT-2` and `Llama2/3` models. In the absolute embedding case, the same value is added to the token embedding for each token located in the same position in the parallel representation of the input text. Likewise in RoPE, absolute positional information is the same for two tokens in the same "location" within parallel sequences, and a token within a parallel sequence retains relative positional dependency with other tokens within the same parallel sequence.

## 3.3 Methodology and Theoretical Guarantees

Consider the Attention block with positional encoding (Ferrando et al., 2024; Kobayashi et al., 2021; Su, Ahmed, et al., 2024) for single-headed attention[3], the attention mechanism maps multiple vectors into a single vector in a differential way. We start with a fixed query vector $\boldsymbol{x}_i$ which we give as an input column vector, $\boldsymbol{X} = [\boldsymbol{x}_1, \ldots, \boldsymbol{x}_n] \in \mathbb{R}^{d \times n}$ is the sequence of all $n$ inputs that will be mapped to a single vector, and $\boldsymbol{M}$ is the attention mask, an $n \times n$ lower triangular matrix. The attention operator is defined as follows.

---

[3]We drop the bias, heads and skip connections for brevity, see appendix for the complete formulation

$$\text{ATTN}(\boldsymbol{x}_i, \boldsymbol{X}, \boldsymbol{M}) = \sum_{\boldsymbol{x}_j \in \boldsymbol{X}} M_{i,j} \alpha_{i,j} \left( \boldsymbol{W}_V \left( \boldsymbol{x}_j + \boldsymbol{p}(i,j) \right) \right) \tag{1}$$

$$\boldsymbol{M}_{i,j} := \begin{cases} 1 & \text{if } j \leq i \\ 0 & \text{else} \end{cases} \qquad \alpha_{i,j} := \operatorname*{softmax}_{\boldsymbol{x}_j \in \boldsymbol{X}} \left( \frac{\boldsymbol{x}_i^\top \boldsymbol{W}_Q \left( \boldsymbol{W}_K \left( \boldsymbol{x}_j + \boldsymbol{p}(i,j) \right) \right)}{\sqrt{d}} \right)$$

where $\boldsymbol{W}_Q$ (query), $\boldsymbol{W}_K$ (key), and $\boldsymbol{W}_V$ (value) are $d \times d$ matrices, with $d$ being the internal dimensions of the matrices. $\boldsymbol{p}(i,j)$ is the vector-valued positional encoding for the token at position $j$ relative to position $i$; note that if it is absolute (GPT-2) then $\boldsymbol{p}$ is only a function of $j$, while if relative (RoPE) it is a function of $i - j$.

Recall that $\boldsymbol{x}_i$ is fixed. Equation (1) can therefore be considered to be a weighted sum over all columns of $\boldsymbol{X}$, where the $j$th column is weighted by $\alpha_{i,j}$, and where $\boldsymbol{M}$ prevents future tokens from having any weight.

Since addition is *commutative*, the order in which the individual $\boldsymbol{x}_j$ are provided is lost, *except* as provided by $\boldsymbol{p}(i,j)$, which includes the order information both for the scaling ($\alpha_{i,j}$) and the unscaled vector ($\boldsymbol{x}_j + \boldsymbol{p}(i,j)$). The attention mask $M$ also functionally encodes positional information when attention blocks are applied in series since 'future' $\boldsymbol{x}_j$ are masked out and do not contribute to the sum.

Thus, to prove that Set-Based Prompting is order-independent, it suffices to show that the positional encoding ($\boldsymbol{p}(i,j)$) and attention mask ($\boldsymbol{M}$) are unaffected by re-ordering by an arbitrary permutation of the parallel sub-sequences. Let $S$ be the (unordered!) set of parallel sub-sequences, $S = \{s_1, \ldots, s_\ell\}$ with $\sum_{k=1}^{l} |s_k| = n$. We start by defining the indices on each token (subscript) and which sequences the tokens are members of (superscript), with $\boldsymbol{x}^\emptyset$ indicating a token is a member of no sub-sequences. The tokens before the parallel sub-sequences are treated identically as in the normal case, and we denote them by $\boldsymbol{X}_{start} = \left[ \boldsymbol{x}_1^\emptyset, \ldots, \boldsymbol{x}_{r-1}^\emptyset \right]$ where $r$ is the lowest index in the parallel sub-sequences. The tokens after the parallel sub-sequences can be described similarly $\boldsymbol{X}_{end} = \left[ \boldsymbol{x}_{p+1}^\emptyset, \ldots, \boldsymbol{x}_m^\emptyset \right]$ where $p$ is the greatest index in the parallel sub-sequences and $m$ is the final index. Note that $m = |\boldsymbol{X}_{start}| + (p - r) + |\boldsymbol{X}_{end}|$, where $|\cdot|$ is the length of a sequence of vectors.

Then, for a parallel sub-sequence $s_k$ we say that $s_k = \left[ \boldsymbol{x}_r^k, \ldots, \boldsymbol{x}_q^k \right]$. Note that in our input representation there is no ordering on $s_k \in S$, because we reset the subscripts (positional indices) at the beginning of each sub-sequence. We have simply indexed the sub-sequences for convenience.

By writing it this way we can directly input it into the $\text{ATTN}(\boldsymbol{x}_i, \boldsymbol{X}_s, \boldsymbol{M})$ function with $\boldsymbol{X}_s = \boldsymbol{X}_{start} \circ s_1 \circ \cdots \circ s_\ell \circ \boldsymbol{X}_{end}$, where $\circ$ is concatenation. If we do this we will obtain order independence since $\boldsymbol{p}(i,j)$ and $\boldsymbol{M}_{i,j}$ only use the indices of the tokens, so they are unaffected by ordering of the sub-sequences.

To show how $\boldsymbol{p}(i,j)$ and $\boldsymbol{M}_{i,j}$ (the unmodified $\boldsymbol{M}$) are order independent we need to consider the three possible cases for the input vector $\boldsymbol{x}_i$: 1) $\boldsymbol{x}_i$ is before the sub-sequences, 2) $\boldsymbol{x}_i$ is in the sub-sequences, and 3) $\boldsymbol{x}_i$ is after the sub-sequences. The **first case** is straightforward since $\boldsymbol{x}_i^\emptyset \in \boldsymbol{X}_{start}$. The output is unaffected by the ordering of $S$ as all sub-sequence tokens are masked out by $\boldsymbol{M}_{i,j}$; $i < r$.

The **second case** is where $\boldsymbol{x}_i^k \in s_k$ with $r \leq i \leq p$. In this case if we naively evaluate $\text{ATTN}(\boldsymbol{x}_i^k, \boldsymbol{X}_s, \boldsymbol{M})$ we will get a result that is unaffected by the ordering of $S$, but the activations will be further from the training distribution than necessary, resulting in lowered performance as discussed in Section 4.3. Note that for any $\boldsymbol{x}_j^k \in \boldsymbol{X}_s$, $\boldsymbol{p}(i,j)$ and $\boldsymbol{M}(i,j)$ are only affected by the positional index $j$ (instead of $k$).

Finally, in the **third case** where $\boldsymbol{x}_i^\emptyset \in \boldsymbol{X}_{end}$, the same argument as the second case applies. The input vector $\boldsymbol{x}_i^\emptyset$ can 'see' all previous tokens, but their positional indexing is unchanged under re-orderings of $S$. Thus in all three cases we have sub-sequence order independence.

### 3.3.1 Attention Mask

While the above method is sufficient for sub-sequence order independence, if used in practice the generated representations encountered by the learned weight matrices ($\boldsymbol{W}_V$) and ($\boldsymbol{W}_K$) will be far from the training distribution in both case 2 and case 3 encounter. In case 2 this is due to input vectors attending to all sub-sequences' inputs, *e.g.* if there were three sub-sequences starting at index $i = 2$ the first token of sub-sequence 1 ($\boldsymbol{x}_2^1$) would have non-zero weight on 4 tokens, three of which would have the positional encoding of 2. In training, the model would never see multiple weights with the same positional encoding particularly multiple positional encodings that match the input token. Note that when used in an LLM ATTN() is applied sequentially, so even minor errors can be amplified. For case 3, the results are less severe since the positional encoding multiplicity only occurs for a set of previous tokens, it does not occur for the input token.

We can mitigate some of the issues by modifying the attention mask $\boldsymbol{M}(i, j)$, making it so that case 2 tokens do not encounter out of distribution inputs. To do this we define a new attention mask $\boldsymbol{M}_{i,j}^{k,f}$ that also takes in the sub-sequences index each along with positional index of each token ($\boldsymbol{x}_i^k$, $\boldsymbol{x}_j^f$, etc), while still retaining the sub-sequence order independence we want. With this notation, when $k = f \neq \emptyset$ we have that $\boldsymbol{x}_i^k$ and $\boldsymbol{x}_j^f$ are in the same parallel sub-sequence.

$$\boldsymbol{M}_{i,j}^{k,f} = \begin{cases} \boldsymbol{M}_{i,j} & \text{if } k = f \\ \boldsymbol{M}_{i,j} & \text{if } k = \emptyset \\ \boldsymbol{M}_{i,j} & \text{if } f = \emptyset \\ 0 & \text{else} \end{cases} = \begin{cases} 1 & \text{if } j \leq i \text{ and } k = f \\ 1 & \text{if } j \leq i \text{ and } k = \emptyset \\ 1 & \text{if } j \leq i \text{ and } f = \emptyset \\ 0 & \text{else} \end{cases} \tag{2}$$

Equation 2 still maintains order independence since it only differs from $\boldsymbol{M}_{i,j}$ when $i \neq j$ and in that case does not depend on the values of $i$ or $j$.

If we consider $\text{ATTN}(\boldsymbol{x}_i^k, \boldsymbol{X}_s, \boldsymbol{M}_{i,j}^{k,f})$ we still retain order independence because $\text{ATTN}(\boldsymbol{x}_i^k, \boldsymbol{X}_s, \boldsymbol{M}_{i,j}^{k,f})$ does not change under re-orderings of the labels, it only considers equality or absence of labels. See Section 4.3 for results when we don't do this masking.

This modified attention mask means that only tokens in case 3 see out of training distribution representations, and that no tokens will 'see' multiple other tokens at the same index.

This yields the following theorem.

**Theorem 1** *Given $\boldsymbol{M}_{i,j}^{k,f}$ as in Equation (2), fix any permutation function $\tau$ on the indices $1, \ldots, \ell$ of the sub-queries $S = \{s_1, \ldots, s_k, \ldots, s_\ell\}$ for the attention mechanism, so that applying $\tau$ to the blocks of column vectors corresponding to the $\ell$-th parallel sub-sequences transforms $\boldsymbol{X}_s = \boldsymbol{X}_{start} \circ \{[\boldsymbol{x}_1^1, ...], \ldots, [\boldsymbol{x}_1^k, ...], \ldots, [\boldsymbol{x}_1^\ell, ...]\} \circ \boldsymbol{X}_{end}$ to $\boldsymbol{X}_s' = \boldsymbol{X}_{start} \circ \{[\boldsymbol{x}_1^{\tau(1)}, ...], \ldots, [\boldsymbol{x}_1^{\tau(k)}, ...], \ldots, [\boldsymbol{x}_1^{\tau(\ell)}, ...]\} \circ \boldsymbol{X}_{end}$. Then*

$$\text{ATTN}(\boldsymbol{x}_i^k, \boldsymbol{X}_s, \boldsymbol{M}_{i,j}^{k,f}) = \text{ATTN}(\boldsymbol{x}_i^{\tau(k)}, \boldsymbol{X}_s', \boldsymbol{M}_{i,j}^{k,f}) \tag{3}$$

*and*

$$\text{ATTN}(\boldsymbol{x}_i^\emptyset, \boldsymbol{X}_s, \boldsymbol{M}_{i,j}^{k,f}) = \text{ATTN}(\boldsymbol{x}_i^\emptyset, \boldsymbol{X}_s', \boldsymbol{M}_{i,j}^{k,f}). \tag{4}$$

## 4  Performance

While Set-Based Prompting is guaranteed to produce order independent results, we still need to test what other impacts it has on the generated text. To this end, we collected four different LLM families (`GPT-2` (Radford et al., 2019), `Llama 2` (Touvron, Martin, et al., 2023),

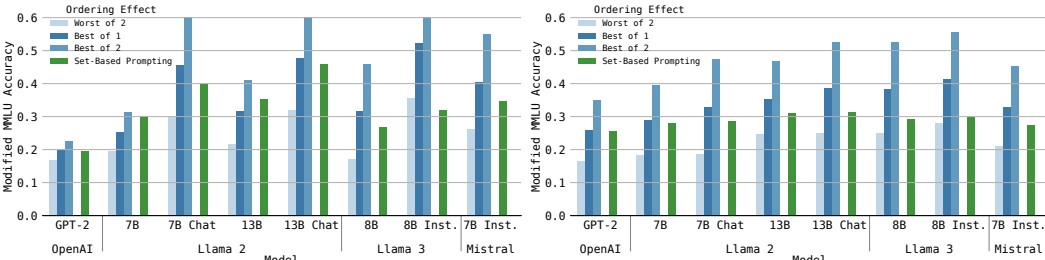

(a) Modified CSQA results for each model      (b) Modified MMLU results for each model

Figure 3: Per model accuracy on two different datasets, blue bars (left three) indicate runs done *without* our method and green with Set-Based Prompting . The blue bars are constructed by running the test twice, once with the normal ordering and once with the reversed ordering. `Worst of 2` and `Best of 2` count when both orderings lead to an correct answer or only one ordering answered correctly, respectively. While `Best of 1` indicates that the normal ordering led to correct answers. As Set-Based Prompting is invariant to reordering so we only show one bar for all orderings.

`Llama 3` (AI@Meta, 2024), and `Mistral` (Jiang et al., 2023)) to conduct our experiments (see figure 1 for the list of models). Due to our method modifying the internals of the model, each family requires a custom implementation of the inference code, meaning we can test all models in a family, but adding new families is time consuming. Our implementation for inputting parallel sub-sequences is implemented like the special tokens already commonly used by most LLMs, with three special 'tokens' (`start-parallel`, `new-sub-sequence`, and `end-parallel`) that are removed during inference, allowing for Set-Based Prompting to be added in existing workflows directly.

For testing we used two standard test sets CommonsenseQA (CSQA) (Talmor et al., 2019) and (Measuring Massive Multitask Language Understanding) (MMLU) (Hendrycks et al., 2020); these were chosen because they are multiple choice question sets that allow us to use the multiple options as the parallel sub-sequences.

As we want the runs with and without Set-Based Prompting to be similar we modified the prompting for both datasets. Instead of using numbers or letters before each option we quote the options in double quotes (") and separate them with a space. This allows for the parallel queries to have the exact same set of tokens as the non-parallel queries. Of note, we implement the normal 1-shot evaluation for these tests that looks at the probability of each answer being generated, and we use the greatest value as the selected answer. We do observe that this slightly reduces the accuracy of the models from prompting with numbers or letters, so we label the methods 'modified CSQA' and 'modified MMLU' to differentiate our results. As we are concerned with the relative performance of our method, not in benchmarking the models, we believe this gives us results that are close to the existing literature.

## 4.1 CommonSenseQA

As we are concerned with reducing the variation of models under reordering, we examine both the accuracy under the default ordering, and the potential accuracy under reordering. In Figure 3a we examine our model's accuracy on CSQA, both with and without Set-Based Prompting . For the order dependent results we run CSQA twice and divide the answers into 4 sets (as in a confusion matrix): questions that both orderings get correct are counted and added to the `Worst of 2` count, while questions that only one gets correct are also added to the `Best of 2` bar, with the ones the normal ordering got correct used for `Best of 1`. The counts are then divided by the total number of questions (9741) to give the mean accuracy. Thus, the range of the blue bars can be considered to be the possible range of accuracies under two possible orderings per question. Note that if the models understood the text the difference between orderings would be minuscule; the difference between just two orderings being large shows that the specific models are fundamentally incapable of solving multiple choice questions, they merely approximate a good response in expectation.

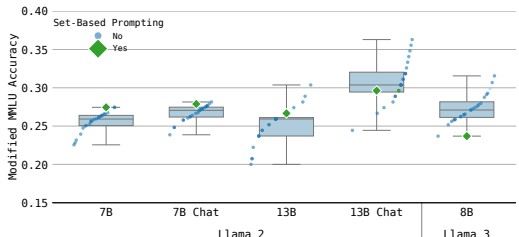
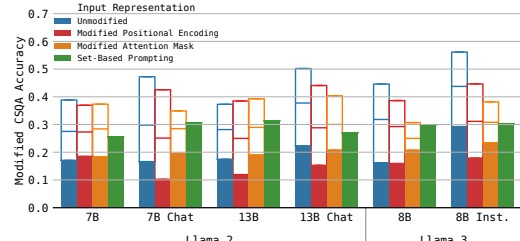

Figure 4: MMLU results for a subset of models across all possible permutations (4!) of the ordering of the options, with the accuracy under Set-Based Prompting indicated with a diamond. Dots are ordered by accuracy with in each model's results, boxes show the quartiles across the different ordering.

Figure 5: Accuracy per model on MMLU, with error bars showing the variation in accuracy under two orderings. The conditions are unmodified model, only the positional encoding $p(i, j)$ modified, only the attention mask $M_{i,j}^{k,f}$ modified, and Set-Based Prompting

When we examine our method we see (1) that all ordering produce the same result, and (2) that the accuracy is within that of the variation of order dependent queries for all models except `Llama 3, 8B Instruct`. This result suggest that Set-Based Prompting is not adding additional information into the model, instead it is removing a dimension of variation. We discuss some hypotheses for how our method impacts the response generation in section 5.

## 4.2 Measuring Massive Multitask Language Understanding

Figure 3b shows the same analysis as in section 4.1 but for MMLU (Hendrycks et al., 2020). We see similar results to CSQA, but with lower accuracy on the more complex MMLU.

To further explore the impact of reordering on outputs we ran all possible permutations of MMLU option orderings through a subset of our models. In Figure 4 we see that for all `Llama 2` models Set-Based Prompting is above or within the inter-quartile range of the 24 samples. We do see that in `Llama 3` our method is as bad as the worst ordering, but it is still within the range of orderings.

## 4.3 Ablations

**Partial Implementations.** While we have a proof as to why both the the positional encoding and attention mask need to be modified, the transformer models are large enough that empirical observations are wanted to confirm the theory. We tested variations of Set-Based Prompting where only the attention mask or the positional encoding was modified to mask the positional information. Figure 5 shows the variation between normal ordering and reverse ordering of question options for the first 20 sets of questions in MMLU (ordered alphabetically). Interestingly, modifying the attention mask reduces variation much more than the positional encoding, although the effect varies significantly between models.

**Enumerated Options.** We ran a subset (first 20) of MMLU with the option numbers re-added, this lead to an improvement on all orderings. Note that adding numbers to inputs encoded with Set-Based Prompting implicitly removes the guarantee of order independence as there is an ordering implied by the numbers. See figure 10 in the appendix for this result.

**Chain of Thought.** We implemented a simple chain of thought prompt ("A: Let's think step by step") for the first 20 MMLU question sets. This lets us determine if the effects of Set-Based Prompting are limited to inputs near the end of the input sequence and check if an alternative prompting method would change the results. This method produced a moderate uplift in performance for both order dependent and order independent cases, note that this experiment was run once and not tuned. See figure 11 in the appendix for this result.

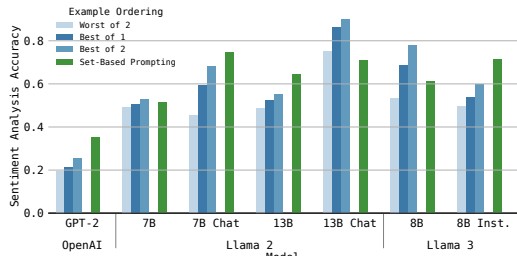 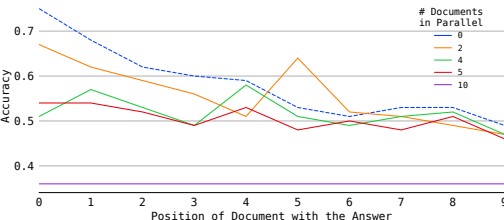

(a) Sentiment classification task with in-context learning. The model was provided with 4 samples with labels, then asked to classify the fifth. Samples orderings are changed betwen bars.

(b) Accuracy on extracting a pieces of information from 10 documents. 0 parallel batches is the fully order dependent model, with others being different splitting locations.

## 4.4 Other Tests

**In Context Learning (Sentiment Analysis).** We implemented a sentiment classification task with in context learning. The model was provided with 4 samples with labels, then asked to classify a fifth. The dataset is *Financial Phrase Bank* Malo et al., 2014 so all statements are finance related and classification is positive or negative. To look at the effects of ordering on the model performance we always had three samples with the same label and 1 with the other label, with the different label always being first or last. See figure 6a for the impacts of different orderings on the final classification task accuracy. Our experiment shows that Set-Based Prompting applied to the examples often improves performance over even the `Best of 2` case.

**Long-Context Information Retrieval.** We implemented the long-context information retrieval task described in the paper by *Liu et al.* N. F. Liu et al., 2024. To do this, we generated 10 document sequences with the answer hidden in one of them. Then we moved where the answer was located from the first to the last document and observed the retrieval accuracy. We used the same templates and dataset as *Liu et al.* for this. To test the effects of Set-Based Prompting we ran the documents in parallel, either all 10, 5 groups (2,2,2,2), four groups (3,3,3,1) or two groups (5,5). When running the sets of documents in parallel there are two opposing forces affecting the accuracy: (1) parallelism, naturally, reduces order-dependence, which helps accuracy; (2) at the same time, the intervention moves the inputs farther out of distribution, reducing accuracy. Figure 6b shows the results of this experiment. Our experiment suggests that limited intervention is a sort of 'sweet spot'. The existence of a 'sweet spot' suggests that our method can be used to evaluate the robustness of the model's learned manifold, since we now have a subtle measure of model sensitivity to perturbations.

**Extended Context Window.** We checked if our method allows for the context window to be *functionally* extended since $n$, the total number of tokens, is less than $m$, the maximum positional index. Text generated where $n$ was greater than the context window of the model maintains coherence when run, see the appendix section 6.12 for an example.

## 5 Discussion

Understanding why Set-Based Prompting works at all, without needing additional training, will require further research. We hope that adding Set-Based Prompting during training will reduce the impact on performance. As shown in Section 4, for all tested models Set-Based Prompting does not cause the model performance to degrade to an unusable state, although, as noted in Section 1, order dependency raises significant bias and reliability concerns.

We hypothesize three mechanisms of action for Set-Based Prompting to negatively impact an LLM's performance. First, Set-Based Prompting reduces the total amount of computation, thus reducing the model's ability to 'do work' leading to lower accuracy. Second, Set-Based

Prompting reduces the information available for some tokens, specifically those in the parallel sub-sequence, *i.e.,* when the LLM is processing the second option it can see the first so can make a more nuanced embedding. Finally, Set-Based Prompting leads to out of distribution token embeddings. The out of distribution impact is suggested by the difference in impact on instruction tuned models, compared to their base models. For example, in Figure 3b we see that for `Llama 3`, Set-Based Prompting performance is almost identical between the instruct and base models, while the instruct performs significantly better on best-of-2. This suggests that Set-Based Prompting is moving the input embeddings such that they are 'outside of' the fine-tuning, but the base model is robust enough to still understand them. These modifications to the model's output may also be mitigated by including examples of Set-Based Prompting inputs during training or fine-tuning, but additional work will be needed to determine this. Notably we also see hints of this when examining the model outputs under our methods; Set-Based Prompting makes fine-tuned models appear closer to the non-fine-tuned models, in particular Set-Based Prompting tends to make the fine-tuned models more verbose. We believe that each of these mechanisms can be tested, allowing for more details to be revealed about the inner workings of language models, and hopefully build towards reducing biases and increasing robustness of these systems.

## 5.1 Further Explorations

In this paper we present and evaluate a small change to the input representation. Our results suggest that larger changes may not only be possible, but may not require significant training overhead. We chose not to study fine-tuning as we believe that showing that our method is 'drop-in' is a significantly more powerful result than one that requires additional training. A next step would be to evaluate how to fine-tune models to be more receptive to this method, possibly even including the training as a part of base-model construction. Additionally, we chose to study the smallest possible version of this effect, a single parallel sub-sequence per prompt. Further study is merited to determine if more complex sub-sequence relationships can be input without significant degradation. In particular, we hypothesize that a version of this method that treats the input text as a directed graph of tokens (instead of a linear sequence) would be possible, *i.e.,* allow nested parallel sub-sequences, supporting cycles will likely require training. This is a representation already used for graphs (Ju et al., 2024; Wang et al., 2020) and is the natural representation for transformers. This type of representation would allow for inputs that much more closely represent how humans are provided text, *e.g.,* consider a class's syllabus, it is often a series of unordered sets of articles to read.Developing techniques that allow for reasoning about human behavior to be translatable to LLMs will greatly improve their utility to non-experts.

## 5.2 Towards Metadata in LLM Inputs

The empirical results in section 4.2 show that, while the positional encoding is used by LLMs, they can nonetheless 'understand' inputs where the positional encoding is modified. This suggests that LLMs are generally robust to minor changes in the token embeddings, with the positional encoding being the only currently implemented example. This is an entirely unexplored area of research. If we consider the positional encoding as a learned type of metadata, then other types of metadata could be added to tokens by simply adding new vectors during training to the token embeddings, just as we do with positional encodings. Adding additional metadata to each token would allow for 'out-of-band' signals that are added to input texts. Consider, for example, tagging each token with a vector that encodes the privilege level of the token, allowing for the implementation of instruction hierarchies (Wallace et al., 2024) that are encoded per token, instead of contextually by tags. Another, example of the utility of adding token metadata is in typography. Many digital texts have **bold**, underlines, *italics*, etc.; each of these could have an associated vector during training, allowing the LLM to 'see' the text more clearly. Instruction hierarchies and typography are just two possible uses of complex LLM inputs, and we believe that many more are possible, allowing for LLMs that can interact with the world's details and not just simple linear text.

As the usage of LLMs becomes more systemic, operators will need to be able to encode more complex representations, and have guarantees on the LLMs' behavior. We believe that this work presents a development towards that goal.

## Acknowledgments

This research is funded in part by the *Alfred P. Sloan Foundation*, "Pseudo-Randomness and the Crystal Ball," (with O. Reingold), G-2020-13941, 2020, and NSF CAREER DMS-2340241.

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

## 6 Appendix

### 6.1 Code Release

Full code for our experiments is available as a part of the final paper release. The code is released under the MIT license. The code can be found at this URL: https://github.com/reidmcy/set-based-prompting.

### 6.2 Compute Resources

Most of our results were done on a single Nvidia A100 GPU with 80GB of memory, running one model, thus the lack of 70B model results. Each run for table 3b took about 12 hours per model, with larger models taking slightly longer.

### 6.3 Computational Complexity

The computational complexity of our method is slightly lower than that of the normal attention mechanism. This is due to removal of elements from the attention mask, which reduces the number of key-value computations. This reduction is thus proportional to the number of parallel sequences and the length of these sequences. The reduction is thus minor for most uses, but could be significant for large numbers of parallel sequences or very long subsequences.

### 6.4 Full Attention Formulation

The formulation of attention presented in section 3.3 has the relevant components for this work, but does not represent the attention mechanism as used in most language models, since the development of multi-headed attention (Vaswani et al., 2017). Again this version is based on that presented in Kobayashi et al., 2021 with the attention mask made explicit and the vectors reordered to make the types work.

The attention mechanism with multiple heads, and bias is:

$$\text{multi-ATTN}(\boldsymbol{x}_i, \boldsymbol{X}, \boldsymbol{M}) = \text{ATTN}^1(\boldsymbol{x}_i, \boldsymbol{X}, \boldsymbol{M}) \circ \text{ATTN}^2(\boldsymbol{x}_i, \boldsymbol{X}, \boldsymbol{M}) \circ \dots \tag{5}$$

$$\text{ATTN}^h(\boldsymbol{x}_i, \boldsymbol{X}, \boldsymbol{M}) := \sum_{\boldsymbol{x}_j \in \boldsymbol{X}} \boldsymbol{M}_{i,j} \alpha_{i,j} \left( \boldsymbol{W}_V^h \boldsymbol{b}_V^h \left( \boldsymbol{x}_j + \boldsymbol{p}(i,j) \right) \right) \tag{6}$$

$$\boldsymbol{M}_{i,j} := \begin{cases} 1 & \text{if } j \leq i \\ 0 & \text{else} \end{cases} \tag{7}$$

$$\alpha_{i,j}^h := \operatorname*{softmax}_{\boldsymbol{x}_j \in \boldsymbol{X}} \left( \frac{\boldsymbol{x}_i^\top \boldsymbol{W}_Q^h \boldsymbol{b}_Q^h \left( \boldsymbol{W}_K^h \boldsymbol{b}_K^h \left( \boldsymbol{x}_j + \boldsymbol{p}(i,j) \right) \right)}{\sqrt{d}} \right)$$

where the superscript $h$ gives the index of the head, $\circ$ is the concatenation operator, and $\boldsymbol{b}_Q^h$ (query), $\boldsymbol{b}_K^h$ (key), and $\boldsymbol{b}_V^h$ (value) are the bias terms for each head. Note also that when used in a model the attention mechanism is likely to be used with a skip connection

$$\boldsymbol{y} = \text{multi-ATTN}(\boldsymbol{x}_i, \boldsymbol{X}, \boldsymbol{M}) + \boldsymbol{x}_i \tag{8}$$

and/or normalization. None of these listed changes affect the proof of presented in section 3.3, with the possible exception of the normalization which could in theory include positional information, but in all four model families we evaluated did not.

We also do not consider sparse attention mask patterns (Parmar et al., 2018), some of which are dependant on the token values (Kitaev et al., 2020).

Table 1: Models used in this analysis divided by family.

| Organization | Model Name | Parameters (B) | Fine-tuned | Base Model | |
|---|---|---|---|---|---|
| OpenAI | GPT-2 | 1.5 | No | | 🤗 |
| Meta | Llama-2-7b | 7 | No | | 🤗 |
| Meta | Llama-2-7b-chat | 7 | Yes | Llama-2-7b | 🤗 |
| Meta | Llama-2-13b | 13 | No | | 🤗 |
| Meta | Llama-2-13b-chat | 13 | Yes | Llama-2-13b | 🤗 |
| Meta | Meta-Llama-3-8B | 8 | No | | 🤗 |
| Meta | Meta-Llama-3-8B-Instruct | 8 | Yes | Meta-Llama-3-8B | 🤗 |
| Mistral | Mistral-7B-Instruct-v0.2 | 7 | yes | Mistral-7B-v0.2 | 🤗 |

## 6.5 Model Details

## 6.6 Alternative Proof

Below is an alternative proof of the main result using a different method. Note that the notation slightly differs from that used in the main proof

Consider the Attention block with positional encoding (Ferrando et al., 2024; Kobayashi et al., 2021; Su, Ahmed, et al., 2024) for multi-headed attention[4]. We take a sequence of $n$ input vectors $\{\boldsymbol{x}_i\}_{i=1}^n \subset \mathbb{R}^D$, written compactly as an input matrix $\boldsymbol{X} = [\boldsymbol{x}_1; \ldots; \boldsymbol{x}_n] \in \mathbb{R}^{D \times n}$, where each $\boldsymbol{x}_i$ is a column of $\boldsymbol{X}$ (also a token). For an input feature $\boldsymbol{x}_j$, we denote the feature matrix after incorporating the positional encoding matrix as $\boldsymbol{X}^{P,j} = [\boldsymbol{x}_1^{p(j,1)}; \ldots; \boldsymbol{x}_n^{p(j,n)}]$.

The self-attention layer with $L$ heads, attention mask matrix $\boldsymbol{M}$, and parameters $\theta = \{\boldsymbol{M}, \{(\boldsymbol{Q}_m, \boldsymbol{K}_m, \boldsymbol{V}_m)\}_{m=1}^L\}$ (each $\boldsymbol{M}$, $\boldsymbol{Q}_m$, $\boldsymbol{K}_m$ and $\boldsymbol{V}_m$ are $D \times D$ matrices) takes $\boldsymbol{X}$ as input and outputs a $D \times n$ matrix. In vector form, the $j$-th column is

$$\text{Attn}_\theta(\boldsymbol{X})_j := \boldsymbol{x}_j + \sum_{m=1}^L \sum_{j' \in [n]} M_{j,j'} \sigma\left(\langle \boldsymbol{Q}_m \boldsymbol{x}_j, \boldsymbol{K}_m \boldsymbol{x}_{j'}^{p(j,j')}\rangle\right) \boldsymbol{V}_m \boldsymbol{x}_{j'}^{p(j,j')}, \quad j \in [n], \quad (9)$$

where $\sigma$ is the activation function, usually taken as the soft-max function.

In the standard decoder-only transformers, $\boldsymbol{M}$ is a lower triangle matrix with $M_{j,j'} = 1$ for all $j \geq j'$, and 0 otherwise. Such an $\boldsymbol{M}$ functionally encodes positional information when attention blocks are done in series since "future" $\boldsymbol{x}_j$ are masked out and do not contribute to the computation. In this case

$$\text{Attn}_\theta(\boldsymbol{X})_j := \boldsymbol{x}_j + \sum_{m=1}^M \sum_{j'=1}^j \sigma(\langle \boldsymbol{Q}_m \boldsymbol{x}_j, \boldsymbol{K}_m \boldsymbol{x}_{j'}^{p(j,j')}\rangle) \boldsymbol{V}_m \boldsymbol{x}_{j'}^{p(j,j')}, \quad j \in [n],$$

which computes $\text{Attn}_\theta(\boldsymbol{X})_j$ based on the weighted summation of all previous tokens $\boldsymbol{x}_{j'}$ for $j' \in \{1, 2, \ldots, j\}$, with weights given by $\sigma(\langle \boldsymbol{Q}_m \boldsymbol{x}_j, \boldsymbol{K}_m \boldsymbol{x}_{j'}^{p(j,j')}\rangle)$.

We now propose to modify the matrix $M$ to make the transformers order independent. Specifically, let us first assume there are $l$ sub-sequences, with indices sets $b_0, b_1, \ldots, b_l, b_{l+1}$ such that $b_0 \cup b_1 \cup \ldots \cup b_{l+1} = [n]$. Suppose we now want the model invariant with respect to the order of $b_1, \ldots, b_l$, that is, we want the $\text{Attn}\theta(\boldsymbol{X})$ output to be unaffected by re-orderings of $b_1, \ldots, b_l$ in $\boldsymbol{X}$.

To achieve this, our proposed Multidimensional Attention Mask sets $M_{j,j'} = 0$ for all $j \in b_1 \cup \ldots \cup b_l$ and $j' \notin b_{\lambda(j)}$ and $j' \leq j$, where $\lambda : [n] \to \{0, 1, \ldots, l+1\}$ is the membership function such that $\lambda(j) = k$ if and only if $j \in b_k$. This makes sure that each sub-sequence cannot see other sub-sequences. In addition, we take positional encodings $p(j, j')$ that depend

---

[4]We drop the bias and output for brevity, see appendix for the complete formulation

only on the relative order between $i$ and the rank of $j$ within $b_{\lambda(j)}$. As a result, the positional encoding for $\boldsymbol{x}_j$ only depends on its rank within the subsequence $\boldsymbol{x}_{b_j}$, and does not rely on other subsequences at all.

Let $\tau : \mathbb{R}^{D \times n} \to \mathbb{R}^{D \times n}$ be an arbitrary permutation function that permutes the column blocks in $\boldsymbol{X}$ indexed by $\{b_1, b_2, ..., b_l\}$. We have the following theorem.

**Theorem 2** *Suppose we define $\boldsymbol{M}$ as above. For any permutation function $\tau$ that permutes the columns blocks in $X$ indexed by $\{b_1, b_2, ..., b_l\}$. $\text{Attn}_\theta(\boldsymbol{X})$ defined in* (9) *will be invariant to this order change:*

$$\tau(\text{Attn}_\theta(\boldsymbol{X})) = \text{Attn}_\theta(\tau(\boldsymbol{X})).$$

*This implies that for any $j \in b_{l+1}$,*

$$\text{Attn}_\theta(\boldsymbol{X})_j = \text{Attn}_\theta(\tau(\boldsymbol{X}))_j$$

The proof of Theorem 2 can be found in Section 6.7 below. This theorem shows that by re-indexing the positional encoding and modifying the attention mask, the output of the attention block becomes independent of the ordering of the sub-sequences.

## 6.7 Additional Proof Details

***Proof of Proposition 2:***

As any permutation can be written as the composition of pairwise permutations, in the following, we prove the result for pairwise permutations. More specifically, suppose we consider a $\tau$ that permutes the columns indexed by $b_k$ and $b_{\tilde{k}}$ in $\boldsymbol{X}$, and obtain $\tilde{\boldsymbol{X}}$.

Recall that $\boldsymbol{M}$ is defined by letting $M_{j,j'} = 0$ for all $j' \notin b_{\lambda(j)}$ and $j' \leq j$. We then have for $j \in [n]$

$$\text{Attn}_\theta(\boldsymbol{X})_j = \boldsymbol{x}_j + \sum_{m=1}^{L} \sum_{j' \in [n]} M_{j,j'} \sigma(\langle \boldsymbol{Q}_m \boldsymbol{x}_j, \boldsymbol{K}_m \boldsymbol{x}_{j'}^{p(j,j')} \rangle) \boldsymbol{V}_m \boldsymbol{x}_{j'}^{p(j,j')}$$

$$= \boldsymbol{x}_j + \sum_{m=1}^{L} \sum_{j' \in \{j' \leq j : j' \in b_0 \cup b_{\lambda(j)}\}} \sigma(\langle \boldsymbol{Q}_m \boldsymbol{x}_j, \boldsymbol{K}_m \boldsymbol{x}_{j'}^{p(j,j')} \rangle) \boldsymbol{V}_m \boldsymbol{x}_{j'}^{p(j,j')}.$$

In the following, we discuss three cases separately: $j \in b_k \cup b_{\tilde{k}}$, $j \in b_0 \cup b_{l+1}$, and $j \notin b_0 \cup b_k \cup b_{\tilde{k}} \cup b_{l+1}$.

Case 1: $j \in b_k \cup b_{\tilde{k}}$. Without loss of generality, we assume $k < \tilde{k}$. Then when $j \in b_k$,

$$\text{Attn}_\theta(\boldsymbol{X})_j = (\boldsymbol{x}_j + \boldsymbol{p}_j) + \sum_{m=1}^{L} \sum_{j' \in \{j' \leq j : b_0 \cup b_k \cup b_{l+1}\}} \sigma(\langle \boldsymbol{Q}_m (\boldsymbol{x}_j + \boldsymbol{p}_j), \boldsymbol{K}_m \boldsymbol{x}_{j'}^{p(j,j')} \rangle) \boldsymbol{V}_m \boldsymbol{x}_{j'}^{p(j,j')},$$

implying

$$\tau(\text{Attn}_\theta(\boldsymbol{X}))_j = \boldsymbol{x}_{\tau(j)} + \boldsymbol{p}_{\tau(j)} + \sum_{m=1}^{L} \sum_{j' \in \{j' \leq j : b_0 \cup b_{\tilde{k}} \cup b_{l+1}\}} \sigma(\langle \boldsymbol{Q}_m \boldsymbol{x}_j, \boldsymbol{K}_m \boldsymbol{x}_{j'}^{p(j,j')} \rangle) \boldsymbol{V}_m \boldsymbol{x}_{j'}^{p(j,j')},$$

and the right-hand side equals to $\text{Attn}_\theta(\tau(\boldsymbol{X}))_j$.

When $j \in b_{\tilde{k}}$,

$$\text{Attn}_\theta(\boldsymbol{X})_j = \boldsymbol{x}_j + \sum_{m=1}^{L} \sum_{j' \in \{j' \leq j : b_0 \cup b_k \cup b_{\tilde{k}} \cup b_{l+1}\}} \sigma(\langle \boldsymbol{Q}_m \boldsymbol{x}_j, \boldsymbol{K}_m \boldsymbol{x}_{j'}^{p(j,j')} \rangle) \boldsymbol{V}_m \boldsymbol{x}_{j'}^{p(j,j')},$$

implying

$$\tau(\text{Attn}_\theta(\boldsymbol{X}))_j = \boldsymbol{x}_{\tau(j)} + \boldsymbol{p}_{\tau(j)} + \sum_{m=1}^{L} \sum_{j' \in \{j' \leq j : b_0 \cup b_k \cup b_{\tilde{k}} \cup b_{l+1}\}} \sigma(\langle \boldsymbol{Q}_m \boldsymbol{x}_j, \boldsymbol{K}_m \boldsymbol{x}_{j'}^{p(j,j')} \rangle) \boldsymbol{V}_m \boldsymbol{x}_{j'}^{p(j,j')},$$

and the right-hand side equals to $\text{Attn}_\theta(\tau(\boldsymbol{X}))_j$.

Case 2: $j \in b_0 \cup b_{l+1}$.

When $j \in b_0$, we have

$$\text{Attn}_\theta(\boldsymbol{X})_j = \boldsymbol{x}_j + \sum_{m=1}^{L} \sum_{j' \in \{j' \leq j: b_0\}} \sigma(\langle \boldsymbol{Q}_m \boldsymbol{x}_j, \boldsymbol{K}_m \boldsymbol{x}_{j'}^{p(j,j')} \rangle) \boldsymbol{V}_m \boldsymbol{x}_{j'}^{p(j,j')}$$
$$= \text{Attn}_\theta(\tau(\boldsymbol{X}))_j.$$

When $j \in b_{l+1}$, we have

$$\text{Attn}_\theta(\boldsymbol{X})_j = \boldsymbol{x}_j + \sum_{m=1}^{L} \sum_{j' \in \{j' \leq j: b_0 \cup b_{l+1}\}} \sigma(\langle \boldsymbol{Q}_m \boldsymbol{x}_j, \boldsymbol{K}_m \boldsymbol{x}_{j'}^{p(j,j')} \rangle) \boldsymbol{V}_m \boldsymbol{x}_{j'}^{p(j,j')}$$
$$= \text{Attn}_\theta(\tau(\boldsymbol{X}))_j.$$

Case 3: $j \notin b_0 \cup b_k \cup b_{\tilde{k}} \cup b_{l+1}$. We have

$$\text{Attn}_\theta(\boldsymbol{X})_j = \boldsymbol{x}_j + \sum_{m=1}^{L} \sum_{j' \in \{j' \leq j: b_0 \cup b_k \cup b_{\tilde{k}}\}} \sigma(\langle \boldsymbol{Q}_m \boldsymbol{x}_j, \boldsymbol{K}_m \boldsymbol{x}_{j'}^{p(j,j')} \rangle) \boldsymbol{V}_m \boldsymbol{x}_{j'}^{p(j,j')},$$

implying

$$\tau(\text{Attn}_\theta(\boldsymbol{X}))_j = \boldsymbol{x}_{\tau(j)} + \boldsymbol{p}_{\tau(j)} + \sum_{m=1}^{L} \sum_{j' \in \{j' \leq j: b_0 \cup b_k \cup b_{\tilde{k}}\}} \sigma(\langle \boldsymbol{Q}_m \boldsymbol{x}_j, \boldsymbol{K}_m \boldsymbol{x}_{j'}^{p(j,j')} \rangle) \boldsymbol{V}_m \boldsymbol{x}_{j'}^{p(j,j')},$$

and the right-hand side equals to $\text{Attn}_\theta(\tau(\boldsymbol{X}))_j$.

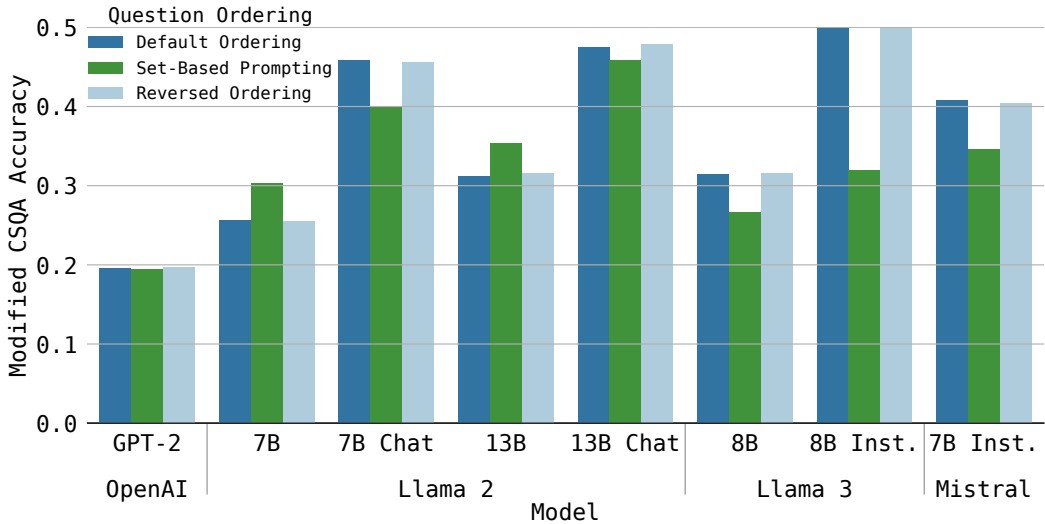

Figure 7: CSQA accuracy for each model with the normal ordering, reversed ordering, and Set-Based Prompting applied to the options.

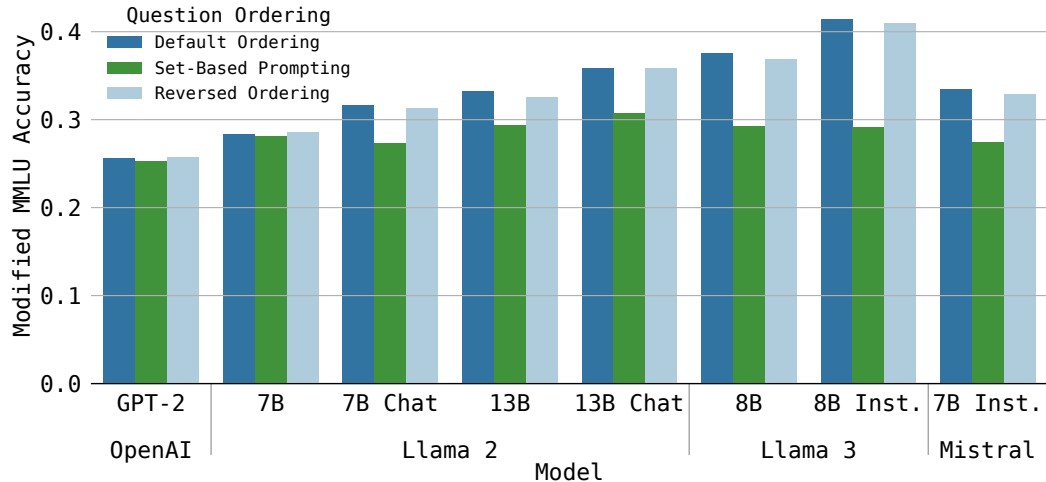

Figure 8: MMLU accuracy for each model with the normal ordering, reversed ordering, and Set-Based Prompting applied to the options.

## 6.8 Additional Plots

Figures 7 and 8 show the same data as 3a and 3b respectively, but with the different orderings accuracy's explicitly included. Figure 9 similarly shows the same data as figure 5.

## 6.9 Impact of Enumeration

Figure 10 shows the effects of adding numbers to the questions during input. Note, this is a removal of order independence for our method as the numbers implicitly encode positioning. We see that this improves performance across the board. We believe this is due to the models being trained with numbers on multiple choice questions.

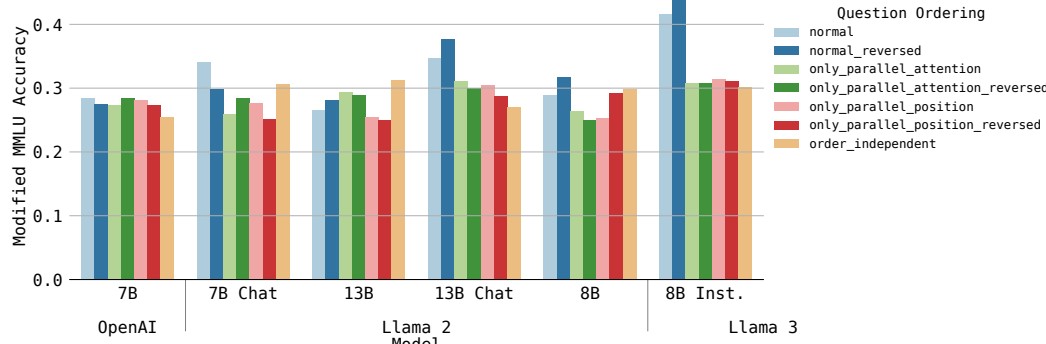

Figure 9: MMLU accuracy for sub set of models with the normal ordering, reversed ordering, accuracy when only the positional encoding $p(i,j)$ is modified, accuracy when only the positional encoding $p(i,j)$ is modified reverse ordering, when only the attention mask $M_{i,j}^{k,f}$ is modified, when only the attention mask $M_{i,j}^{k,f}$ is modified reverse ordering and Set-Based Prompting applied to the options.

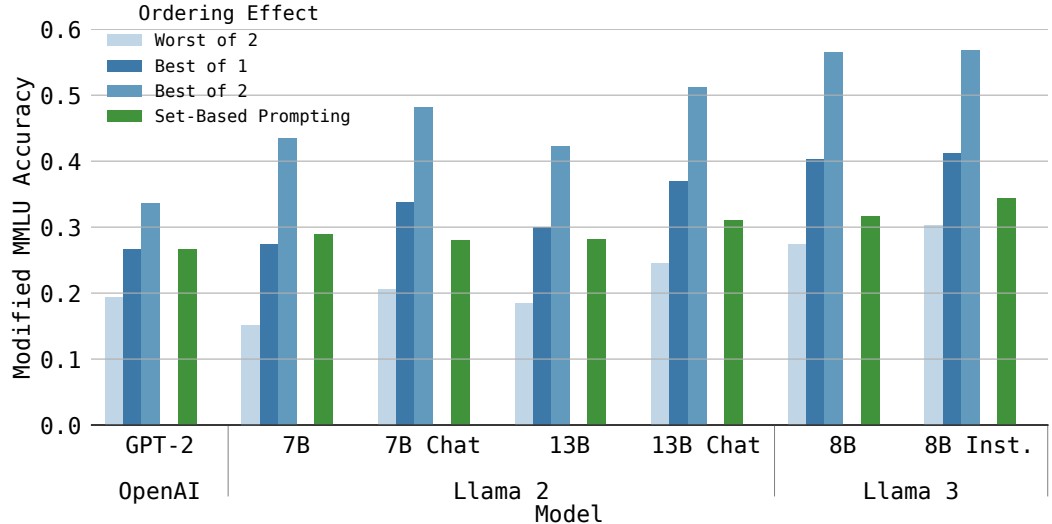

Figure 10: Run of the first 20 MMLU questions (our test set for Figure 4) with the numbers added, formatting identical to figure 3a.

## 6.10   Chain of Thought Prompting

## 6.11   Implementation

We conduct all experimental testing on huggingface models. We perform the greedy decoding with the temperature 0, and apply the zero-shot prompting in all experiments. These huggingface models already accept 2D attention masks as input to the forward() call, which accepts an attention mask and positional encoding as input and outputs a single generated output token. This 2D attention mask support is detailed in https://huggingface.co/blog/poeda-tor/4d-masks. We make a few modifications to these model instance objects in order to enable model instances to accept 2D attention masks as input to the generate() call, which accepts an attention mask and positional encoding and outputs a sequence of generated output tokens. The modifications to Llama and Mistral model instances are identical.

A few additional minor modifications are required for GPT-2 model instances. On the GPT-2, Llama, and Mistral model architectures, we override the model instances' func-

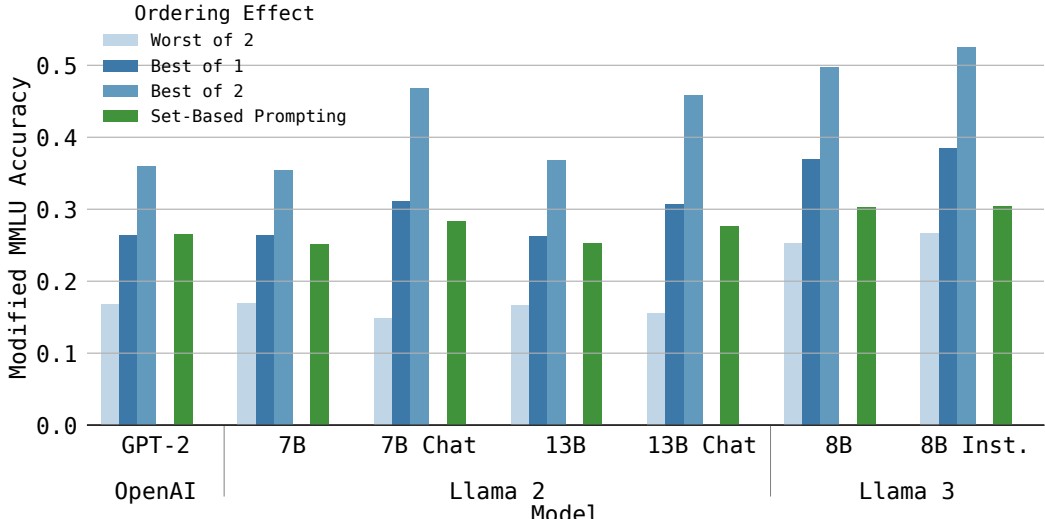

Figure 11: Run of the first 20 MMLU questions (our test set for Figure 4) with chain of thought prompting, formatting identical to figure 3a.

tion _update_model_kwargs_for_generation(). The function implementation remains the same, with the exception that if the function is called with a 2D attention mask, the 2D attention mask is converted to a 1D attention mask to be passed to subsequent forward calls. When generate() is called with a 2D attention mask, it passes that 2D mask to the first forward() call to generate the first token. After each call to forward, the function _update_model_kwargs_for_generation() defines the attention mask and position ids to be passed to the next forward() call. On second or later calls to forward, a 1D attention mask is passed, where attention is paid to all non-padding tokens in the input prompt and all tokens generated thus far. For the llama/mistral models, on subsequent calls, the subsequent position id is defined to consist of a vector containing only the next integer following the largest position id in the previous position id vector.

For example, suppose the input consists of the tokens A,B,C,D, where tokens B and C should be processed in parallel.

We would define the initial attention mask to be

$$\begin{bmatrix} 1 & 0 & 0 & 0 \\ 1 & 1 & 0 & 0 \\ 1 & 0 & 1 & 0 \\ 1 & 1 & 1 & 1 \end{bmatrix}$$

and the initial position ids to be [1,2,2,3].
These values are passed to the first forward() call. On the next call to forward, we pass attention mask [1,1,1,1,1] and [[4]]. On the third call to forward, we pass attention mask [1,1,1,1,1,1] and position ids [[5]], and so on. The second generated token will attend to all original input tokens and the first generated token, and will utilize previously computed keys/values such that it views tokens B and C as though they were still processed in parallel.

On the GPT-2 instances, we override two additional functions. First, we override the '_attn' function of the GPTAttention object in each hidden layer. If a 2D attention mask is passed, then '_attn' skips generating a causal mask and uses the passed attention mask instead. This modification is unnecessary for Llama, which was already modified to pass a 2D attention mask through all the layers. Second, we override the function 'prepare_inputs_for_generation'. By default, this function regenerates position ids if both position ids and an attention mask are passed to generate(). Since we want to be able to pass custom position ids, we modify

the function to allow both the unmodified position ids and attention mask to be passed to subsequent forward calls.

## 6.12  Extending Context Window

One limitation of LLMs is their finite context windows. Existing work tries to circumvent this issue through techniques that require additional pretraining or and fine-tuning. In particular, Munkhdalai et al., 2024 proposes a new attention technique dubbed Infini-attention that scales LLMs to infinitely long inputs with bounded memory. This method, however, requires continual pre-training and fine-tuning, whereas we present Set-Based Prompting as a drop-in method of significantly increasing the context window when subsets of the text can be processed in parallel.

Due to Set-Based Prompting changing the positional encoding, inputs that exceed the trained context window of the model can be used since only the longest sub-sequence contributes to the positional encoding of the final input token. Thus, we can input longer sequences than the model was designed for. We tested this by inputting sequences of 6,000 tokens into Llama 2 7B, and the model was able to correctly summarize the input, while the same input put sequentially produces garbled text. See below for an example.

For a 6018 token input of the Wikipedia article for a recent disaster Llama 7B output:

```
    .  its its its its its its its its its its its its its its its its its
        its its its its its its its its its its its its its its its its its
         its its its its its its its its its its its its its its its its
        its its its its its its its its its its its its its its its its its
         its its its its its its
  its
  its                      . . . .  .  its its its its its its its its its
     its its its its its its its its
. . .                                   .
```

When the text was broken up by header and fed in parallel the output was:

```
On March 26, 2024, the Francis Scott Key Bridge in Baltimore, Maryland
    collapsed after being struck by the container ship Dali. Two people
    were rescued from the water, and six maintenance workers were
    reported missing and presumed dead. The bridge was a continuous
    truss structure that was fracture critical, meaning it had no
    redundancy against failure of any particular part. The collision
    destroyed the southwest main truss pier, causing the central and
    south spans to collapse. The ship's emergency generators provided
    power to the propulsion system, but not to the electrical systems.
    The investigation is being led by the National Transportation Safety
     Board, with support from the Federal Bureau of Investigation, the U.
    S. Coast Guard, and the Maryland Department of Transportation. The
    ship's crew was accounted for, and no serious injuries were reported.
     The disaster has raised questions about the safety of larger ships
    passing
```

Further work should determine whether this drop-in extended context window method comes at the cost of output quality degradation, relative to methods like Infini-attention.

