# OpenReview forum: "Order-Independence Without Fine Tuning"
_NeurIPS.cc/2024/Conference — NeurIPS 2024 poster_

### Official Review · Reviewer_CaKN · 2024-07-12

**Soundness:** 3
**Presentation:** 1
**Contribution:** 1
**Rating:** 5
**Confidence:** 4

**Summary:**

State-of-the-art language models (LMs) are now used to perform tasks (such as, e.g., question answering) with no fine-tuning; these models are fed the question and target options in their context, and their next-token distribution is then used to select an answer from this set of options.
LM outputs, however, are known to vary significantly based on the order in which these options are fed to it.
This paper proposes set-based prompting, a method to make LMs invariant to such option reorderings.
In short, the method feeds all options to a transformer model “in parallel” by modifying their position indexes to start from the same value, and by masking the attention so that options cannot attend to each other.
In practice, this makes the model run slightly out of distribution, as tokens presented to the model after the QA options will be able to attend to multiple tokens with the exact same position (as the multiple options are assigned this equivalent position indexes).
The paper has experiments on two tasks, question answering and language understanding.

**Strengths:**

The paper presents a simple solution to a well known problem in NLP.

The paper runs experiments with a reasonable number of language model families: gpt-2, llama2, llama3, and mistral.

**Weaknesses:**

In my opinion, this paper presents what is a relatively simple solution as being more complex than it is. (To be clear, I think this simplicity is one of the solution’s positive aspects, and not a negative aspect. Its presentation, however, could be considerably simpler.) Further, the proof of order invariance is sold as an important contribution, but it is also relatively straightforward. Since the methodological/theoretical contributions of this paper are relatively minor, though, I would expect it to have more experiments to confirm its efficacy. The paper, however, only presents two experiments, which are not very comprehensive in assessing the method's performance. More details below.

* The paper’s contribution is a simple (yet potentially efficient) method to make LMs invariant to prompt ordering. This solution amounts to reindexing token positions and slightly changing the model’s attention mask. However, the paper takes almost 6 pages to present this relatively simple solution.
* The paper only runs experiments on two tasks, and with one dataset each. This is not a comprehensive evaluation. The paper could run experiments with other tasks. Specifically, analysing the performance of this method on long-context information retrieval, as in Liu et al. (2024) could be interesting.
* In-context learning is another important setting in which models are susceptible to input order variation. However, the paper only focuses its experiments on zero-shot experiments.

Liu et al. (2024). Lost in the middle: How language models use long contexts




------------

Edit after author responses:

I thank the authors for their response, and apologise for only replying to it after the discussion deadline. While I still think that some aspects of the paper could be significantly improved, e.g., simplifying the presentation, the newly added experiments make the paper's evaluation much more comprehensible, so I am raising my scores (from 3 to 5). The experiments showing the performance degradation on the long-context experiment are quite interesting (even if they present negative results), and experiments with ICL show another important setting for which this method can be useful.

**Questions:**

I don't have any specific questions. But as a suggestion, I think the stacked box plots in Figs 3 and 5 are quite hard to read. It would likely be easier to read them if the different conditions were presented as separate bars.

**Limitations:**

The authors mention that their solution will make models run slightly out of distribution. Expanding on this point could be interesting, maybe discussing possible ways to mitigate this issue, such as fine-tuning models.

---

> ### Author Rebuttal · Authors · 2024-08-07
>
> We thank the reviewer for their time in reading our paper and providing some useful questions.
>
> # Relevancy
>
> Order dependency is an important open problem in the LLM NLP literature [1,2,3,4].  Known motivations include increasing model robustness and reducing ordering biases.  To these we add applications from algorithmic fairness. To our knowledge, we are the first to obtain a theoretically validated solution. Moreover, we demonstrate that our solution incurs essentially no cost in accuracy. The robustness problem is a major concern for LLMs and that they may only know the answer in one ordering [5] suggests that the models are not learning useful representations. Having a method that forces the LLM to answer without using ordering means we can better evaluate LLMs which will affect all LLM evaluations.
>
> # Methodological Elegance
>
> That the method seems obvious in hindsight is a strength, not a weakness!  This elegant solution allows for implementation by adding only three new “tokens” to the tokenizer’s dictionary. We will endeavor to increase clarity, and any suggestions for where to focus our attention are enthusiastically welcomed.
>
> # Experimental Scope
>
> Inspired by [1,2,3,4],  we tested our method across multiple LLM implementations with the 57 subject MMLU as our main target. We also spent considerable time and effort examining which components of our intervention were required in practice: our original hypothesis was that we would also need to modify the padding so that the different sub-sequences were aligned, but we found that this was not needed. We also showed that our method allows for expanding the context window of the model (Sec 4.3), but as that follows directly from the positional encoding formulation we did not emphasize this result.
>
> The reviewer suggests two non-multiple choice question tasks. We have implemented both of them and plots showing our results are available in the main response PDF.
>
> ## Long-Context Information Retrieval
>
> We implemented the long-context information retrieval task described in the paper by Liu et al. [6]. To do this, we generated 10 document sequences with the answer hidden in one of them. Then we moved where the answer was located from the first to the last document and observed the retrieval accuracy. We used the same templates and dataset as Liu et al. for this. To test the effects of our method we ran the documents in parallel, either all 10, 5 groups (2,2,2,2), four groups (3,3,3,1) or two groups (5,5). When running the sets of documents in parallel there are two opposing forces affecting the accuracy: (1) parallelism, naturally, reduces order-dependence, which helps accuracy; (2) at the same time, the intervention moves the inputs farther out of distribution, reducing accuracy.  Our plot suggests that limited intervention is a sort of “sweet spot.”  More interestingly, this suggests that our method can be used to evaluate the robustness of the model’s learned manifold.  This is an exciting new direction.  Thanks again!  We will include more details along with the data in the code release in the final paper.
>
> ## In-Context Learning
>
> In response to the reviewer’s suggestions, we implemented a sentiment classification task with in context learning.  The model was provided with 4 samples with labels, then asked to classify the fifth. The dataset is Financial Phrase Bank [7] so all statements are finance related and the model is attempting to classify them as positive or negative. To look at the effects of ordering on the model performance we always had three samples with the same label and 1 with the other label, with the different label always being first or last. When we ran this test, the in-context learning improved over the 1 shot test, showing that the models are using the examples.
>
> We found that putting the examples in parallel often improves the accuracy over the non-parallel case, or is as good. This was a very strong showing by our method and we thank the reviewer for suggesting it! The plot can be seen in the main response PDF. We will include the plot and details of the test in the paper along with releasing the code.
> Bar plots
>
> We have redone them as separate bars, see main response PDF, and will update the paper with the new style for all plots.
>
> # Discussion of fine-tuning
>
> We briefly discussed fine-tuning in section 5.1, but mostly to say we did not examine it. We agree that fine-tuning is the likely best way to mitigate the accuracy degradation. We will add more to the discussion on this. If there are other areas we can improve the discussion on please let us know and we can discuss them during the decision period.
>
> # Citations
>
> [1] Alzahrani, N., Alyahya, H.A., Alnumay, Y., Alrashed, S., Alsubaie, S., Almushaykeh, Y., Mirza, F., Alotaibi, N., Altwairesh, N., Alowisheq, A., Bari, M.S., Khan, H., 2024. When Benchmarks are Targets: Revealing the Sensitivity of Large Language Model Leaderboards.
>
> [2] Chen, X., Chi, R.A., Wang, X., Zhou, D., 2024. Premise Order Matters in Reasoning with Large Language Models. https://doi.org/10.48550/arXiv.2402.08939
>
> [3] Pezeshkpour, P., Hruschka, E., 2023. Large Language Models Sensitivity to The Order of Options in Multiple-Choice Questions. https://doi.org/10.48550/arXiv.2308.11483
>
> [4] Zheng, C., Zhou, H., Meng, F., Zhou, J., Huang, M., 2023. Large Language Models Are Not Robust Multiple Choice Selectors. http://arxiv.org/abs/2309.03882
>
> [5] Oren, Y., Meister, N., Chatterji, N.S., Ladhak, F. and Hashimoto, T., Proving Test Set Contamination in Black-Box Language Models. In The Twelfth International Conference on Learning Representations.
>
> [6] Liu et al. (2024). Lost in the middle: How language models use long contexts
>
> [7] Malo, P., Sinha, A., Takala, P., Korhonen, P.J. and Wallenius, J., 2013. Good debt or bad debt: Detecting semantic orientations in economic texts. CoRR abs/1307.5336. URL: http://arxiv. org/abs/1307.5336.

---

### Official Review · Reviewer_gPKU · 2024-07-12

**Soundness:** 4
**Presentation:** 4
**Contribution:** 4
**Rating:** 9
**Confidence:** 4

**Summary:**

The paper addresses the problem of order dependency in Large Language Models (LLMs), which causes inconsistency in outputs when the order of semantically identical sub-sequences is changed. The authors propose a technique that eliminates order dependency in transformer-based LLMs. The method is both theoretically sound and experimentally validated, showing minimal impact on accuracy despite the inputs being out of distribution. The technique can be seamlessly integrated into fully trained models. The paper also explores the potential of enhancing LLMs by incorporating metadata information similar to positional encoding.

**Strengths:**

- The proposed method elegantly and effectively addresses the order dependency issue in LLMs with only a minor performance drop.
- The method is supported by thorough experimental evaluation and theoretical guarantees, adding to its credibility.
- The paper is well-written, clearly explaining the methodology and its implications.
- Section 5.2 introduces an innovative idea of improving LLMs by using metadata information in a manner akin to positional encoding, which opens up exciting new avenues for research.

**Weaknesses:**

The review does not identify any significant weaknesses in the paper. However, there are a few points of clarification needed, as outlined in the Questions section.

**Questions:**

- What is the impact of not enumerating the candidate answers (e.g., using A, B, C, D) versus simply using “”? This aspect does not appear to be addressed in the main text.
- Regarding Figure 3, the proposed method seems to perform better than the worst-case scenario. However, if the goal is to achieve the highest score, it appears that the order-dependent method might still be preferable. Can the authors elaborate on this?
- Why do the results differ in llama3? A more detailed explanation would help clarify this variation.

This paper presents a significant advancement in addressing order dependency in LLMs. The method is both theoretically and experimentally robust, providing a practical solution with minimal performance trade-offs. The clear writing and innovative approach, particularly in Section 5.2, add to the paper’s strengths. Addressing the questions and clarifying the identified points would further enhance the paper. Overall, this work is a valuable contribution to the field and is strongly recommended for acceptance.

[edit: I increase my rating to Very Strong Accept, see my comment on author's rebuttal for justification]

**Limitations:**

Limitations are adressed in the paper

---

> ### Author Rebuttal · Authors · 2024-08-07
>
> Thank you for the kind review and we are very glad to see other people are excited by the directions this work suggests. As the other reviewers posed many questions, we have some more results to show in the general rebuttal document.
>
> # Question Responses
>
> ## Impact of Enumeration
>
> We didn’t test with the questions numbered in the paper since that leaks positional information. We did some testing to see what quoting methods worked best, notably finding that no quoting has significant degradation across all models and conditions. We re-ran the first 20 MMLU questions (our test set for Figure 4) with the numbers added. This does improve performance across the board. We believe this is due to the models being trained with numbers on multiple choice questions. The order-independent results were within the error bars as in the other tests, but it’s harder to interpret those results as there is positional information leakage. To see the figure please look at the PDF for the main response.
>
> ## Figure 3
>
> Yes, without fine-tuning our method is likely to slightly underperform non-parallel inputs, but as we and others have shown the order dependent  method leads to a significant increase in variance under reordering changes. Our technique reduces variance.
>
> To make the bias reduction concerns more concrete, consider a system that uses an LLM to compare essays. This system will likely be biased towards preferring the earlier essays (thanks to the Lost Middle Phenomenon). So if the essays are provided in alphabetical order the system will likely be biased against Chinese writers due to their names more often being later in the (English) alphabet. Making systems that don’t have these subtle biases is an important part making deployment of LLMs possible in the real world.
>
> ## Llama3
>
> We found the lower performance on Llama3 very interesting too. We hypothesize that this is due to Llama3 being overtrained relative to Llama2, making it more sensitive to out of distribution inputs. Interestingly, qualitatively we observed that our method also tends to make the chat fine-tuned models behave more like their base models (less terse responses was the main quantitative observation), suggesting that the fine-tuning was also partially interfered with by our method, resulting in the model responding like its base under our method. We think this suggests some methods for evaluating the robustness of trained models, and will add some discussion of these results to the main paper.
>
> # Final Thoughts
>
> Thank you for your comments and new experiment suggestion. If there are any more questions we will try to answer them during the discussion period.

---

> > ### Comment · Reviewer_gPKU · 2024-08-09
> >
> > Thank you for the rebuttal and the clarifications to my questions.
> >
> > I must admit that I am surprised by the low ratings given in the other reviews. This paper proposes an elegant and creative method for addressing an important issue observed in LLMs (the authors convincingly explained why order-dependency in language models is problematic). It is supported by thorough experimental evaluation and theoretical guarantees. Moreover, it paves the way for promising applications that extend beyond the specific case of multiple-choice questions, as discussed in Section 5.2 and supported by the additional results for in-context learning provided during this rebuttal.
> >
> > Given these strengths, I believe this paper deserves to be accepted at the conference. Based on the rebuttal, I have decided to increase my initial rating to 9, hoping that this might encourage other reviewers to reconsider their ratings.

---

> > > ### Author Response · Authors · 2024-08-12
> > >
> > > Thank you for the kind response. We agree with your summary of this work.

---

### Official Review · Reviewer_bKgM · 2024-07-16

**Soundness:** 3
**Presentation:** 2
**Contribution:** 2
**Rating:** 5
**Confidence:** 4

**Summary:**

The paper aims to address an issue within large language models (LLMs): their sensitivity to the order of input sequences, known as order dependency. This problem causes LLMs to produce inconsistent outputs when the order of semantically identical inputs is changed.

### Key Contributions:
1. **Set-Based Prompting Technique**: The authors propose a novel technique called Set-Based Prompting, which ensures that the output of an LLM is invariant to the order of a specified set of sub-sequences. This method modifies the input representation by removing the order information from the inputs.

2. **Theoretical Guarantees**: The paper provides theoretical proofs demonstrating that Set-Based Prompting eliminates order dependency for any transformer-based LLM.

3. **Empirical Evaluation**: The authors test their method on multiple LLMs, including GPT-2, Llama 2, Llama 3, and Mistral, using tasks like multiple-choice questions from the CommonSenseQA and MMLU datasets. They show that while Set-Based Prompting can slightly impact performance, it generally maintains accuracy within acceptable bounds and eliminates the variation caused by different orderings.
------------------------------------------------------------------------
Thank you for your replies and I raised my score accordingly.

**Strengths:**

### Strengths

- **Theoretical Rigor**: The paper includes rigorous theoretical proofs that validate the effectiveness of Set-Based Prompting in eliminating order dependency.
- **Empirical Validation**: The extensive empirical evaluation on multiple models (GPT-2, Llama 2, Llama 3, and Mistral) on two datasets (CommonSenseQA and MMLU) demonstrates the robustness and applicability of the method.

**Weaknesses:**

### Weaknesses

#### Implementation Complexity
- **Engineering Trick Perception**: The method may be perceived as an engineering workaround rather than a fundamental advancement. It requires users to input sub-sequence information, which can be seen as an additional burden and may limit the method's practical applicability. Providing a more automated or integrated approach could make the technique more user-friendly.

#### Limited Improvement in Performance
- **Marginal Accuracy Gains**: While the method ensures order independence, the actual improvement in model performance (accuracy) is limited. The results show that Set-Based Prompting maintains accuracy within the variation caused by different orderings but does not significantly enhance it. Highlighting specific scenarios where this method offers substantial performance gains could strengthen the paper's impact.

#### Dependency on User Input
- **Manual Sub-Sequence Specification**: The requirement for users to specify sub-sequences manually can be a significant limitation. This dependency on user input reduces the method's usability and scalability, particularly for large datasets or applications where manual specification is impractical. Exploring ways to automate the identification of sub-sequences would be a valuable enhancement.

#### Evaluation Scope
- **Limited Dataset Variety**: The evaluation is primarily conducted on multiple-choice question datasets (CommonSenseQA and MMLU). While these are standard benchmarks, the scope of evaluation could be broadened.

#### Theoretical vs. Practical Benefits
- **Practical Utility of Theoretical Guarantees**: While the theoretical guarantees are robust, the practical benefits might not be as compelling if the accuracy remains within the range of existing variations. Providing more concrete examples or case studies where order independence significantly improves real-world applications could make the practical utility of the method more apparent.

#### Misleading Paper Title
- **Title Naming Concerns**: I do not think that the proposed method is kind of "prompting" paper. It does change the internal implementation of LLMs instead of simply "prompting" them.

**Questions:**

What will be the  benchmark performance under chain of thought prompting of this method compared to default chain-of-thought prompting baseline?

**Limitations:**

See more details in Weaknesses section.

---

> ### Author Rebuttal · Authors · 2024-08-07
>
> Thank you for taking the time to go over the paper and provide some useful feedback.
>
> # Relevancy
>
> Order dependency is an important open problem in the LLM NLP literature [1,2,3,4].  Known motivations include increasing model robustness [4] and reducing ordering biases [2].  To these we add applications from algorithmic fairness. To our knowledge, we are the first to obtain a theoretically rigorous solution.  Moreover, we demonstrate that our solution incurs essentially no cost in accuracy. The robustness problem is a major concern for LLMs, and that they may only know the answer in one ordering [5] suggests that the models are not learning semantically useful representations. Having a method that forces the LLM to answer without using ordering means we can better evaluate LLMs.
>
> To make the fairness concerns more concrete, consider a system that uses an LLM to compare essays. This system will likely be biased towards preferring the earlier essays (thanks to the Lost Middle Phenomenon). So if the essays are provided in alphabetical order the system will likely be biased against Chinese writers due to their names more often being later in the (English) alphabet. Making systems that don’t have these subtle biases is an important part making deployment of LLMs possible in the real world.
>
> # Additional Comments
>
> The reviewer raises many other points in their review and we address them here.
>
> ## Engineering Trick Perception
>
> Most methods that interact with LLM internals are simple engineering tricks, since the systems are complex, and simple linear methods mean the interventions are well understood. For example, the recent Golden Gate Claude [6] is implemented as a simple linear addition to the model’s activations, and the added vector is learned via a single hidden layer autoencoder. More generally, linear probes are very common in the explainable AI literature and many works have been published about them.  Thus, there is a strong tradition in deep learning using simple methods to enhance precision. Our technique, too, is a simple intervention; moreover, it enjoys a rigorous correctness proof.
>
> See Sub-Sequence Specification below for discussion of how our method can be easily integrated into existing workflows.
>
> ## Accuracy Gains
>
> We do not expect this method to improve accuracy. Our goal is reducing variance and information leakage. We are not providing additional information or additional “thinking time”, we are removing a cognitive defect from the LLM. The results shown in this paper greatly exceeded our hopes, it is very rare to find an intervention that doesn’t completely destroy the model’s performance. So we were hoping to get near the bottom of the error bars, not even fully within the range for all models. We think that this method having such a minor impact is in and of itself something that the community should be aware of, as we discuss a little bit in section 5, reviewer gPKU commented on this also.
>
> ## Sub-Sequence Specification
>
> We have implemented a method to input parallel sub-sequences that works like the special tokens already commonly used by most LLMs. Our system is implemented with three special “tokens”, a start parallel (`<|start_2d|>`), new sub-sequence (`<|split_2d|>`), and an end of parallel (`<|end_2d|>`). During inference our tokenizer does the splitting itself. This should be a simple addition to any tokenizer. The main difficulty for us was verifying there was no accidental information leakage. Making a nicer interface made the analysis much easier, as we simply ignore the parallel tokens during “normal” runs!
>
> The main results in the paper are all done with automated extraction of the parallel sequences, we expect automating the insertion of parallel sub-sequences to be straightforward as many LLM inputs are already constructed via concatenation of strings.
>
> ## Limited Dataset Variety
>
> We also have conducted two non-multiple choice question tasks at the request of the other reviewers. We have implemented a few-shot sentiment analysis task on which our method often outperforms the baseline, and a needle-in-a-haystack test where we match the performance of the baseline. Please see the general response for more details. These two new results will also be added to the paper.
>
> _Please see the global response for more details._
>
> ## Theoretical Guarantees
>
> Theoretical guarantees are few and far between in this exciting literature.  In our case, they lead to elegant solutions to a real-life problem, if the model is not robust to our method then it is likely not robust enough in the real world. As we discuss in the Relevancy section there are many areas where raw performance is traded for lower variance. Note, also that we only look at un-fine tuned models, we expect fine-tuning to significantly reduce the performance gap, if the models are robust.
>
> ## Title Naming Concerns
>
> We agree that the name can be improved! Thank you for pointing this out.  Our current revised candidate name is __Order Independent Input Representations__ . We are happy to discuss this more during the discussion period.
>
> ## Chain of Thought Prompting
>
> In response to the reviewer’s suggestion we implemented a simple chain of thought prompt  (“A: Let's think step by step”) for the first 20 MMLU question sets. We were happy to see that the results match the other MMLU results from the paper, with chain-of-thought providing a moderate uplift on performance for both order dependent and order independent cases! This was our first run with chain of thought, with no tuning. We have added the plot to the general response PDF and will include it in the paper, along with the code and data.
>
> # Final Thoughts
>
> Thank you and if there are any more questions we will try to answer them during the discussion period.
>
> # Citations
>
> _Please see global response_

---

### Author Rebuttal · Authors · 2024-08-07

We thank the reviewers for their thoughtful comments and are glad that everyone appreciates that we have obtained the first rigorously proven solution to an open problem – achieving order-independence – in NLP. Our solution is conceptually intuitive, and while the proof requires careful attention, it aligns well with this intuition.

Attached hereto are the plots associated with the 4 additional experiments the reviewers mentioned. All four show that our method is well within the range suggested by the original experiments, indicating that our method is robust to variations in task as well as in model family. Notably, our method outperforms the best-of-two results for the non-parallel inputs on the few shot learning test.  We provide additional, more tailored, responses to each interviewer individually.

# Relationship to the Literature

The order-dependency problem is well studied in the literature [1,2,3,4]. Prior work focuses on the robustness implications, as a model’s high sensitivity to input orderings mean that tests cannot be trusted to fully represent the model’s underlying understanding of the world. In this paper we add a new motivation for solving this problem: the fairness concerns raised by order dependency  (for example, alphabetical ordering of candidates may lead to discrimination against individuals with Chinese names). Both these concerns are significant to the field; our method is the first to completely solve them, and our solution provably will always work. There has also been further work on the order dependency problem since this paper was submitted, one of the top cited papers out of ICLR this year looked at how order dependency reveals a lack of robustness in models [5], specifically they were interested in looking for training set contamination as the cause.

# Solution Elegance

Our method requires two interventions to the input representation, proving that these have the required effects takes a few pages since we prove our results for the general case of all transformer based LLMs. We took significant effort to make the methods clear and easy to understand and we are happy to see that our results have the property of seeming obvious in hindsight. Our implementation seeks to make our methodology easy to use, and our code release implements the system as a modification to the tokenizer, meaning that we simply add three special tokens (start, break, end) to our texts to make sub-sequences order independent.

# Limited Dataset Variety

Our paper is addressing works highlighting the order dependency problem in multiple choice questions as those are the most common benchmarks for LLMs, so we focused our efforts on providing a broad set of models. Additionally MMLU tests a broad range of tasks (math, law, general knowledge), we reported only the top level numbers due to space constraints, but it is often broken down into the 57 different tests.

We also showed that our method allows for expanding the context window without any additional training (see Sec 4.3), as this paper was primarily focused on addressing the problems on the same terms as the other works in the literature; we only mention this briefly. We will add more details in the final paper on the expanded context window results.

# Figures

The attached PDF has 4 figures. Note that for all figures we will provide code and data release. We focused on the Llama and gpt-2 models for these analyses. If the reviewers have any questions we will seek to address them in the discussion period.

## Fig 1, Chain of Thought Prompting

We implemented a simple chain of thought prompt  (“A: Let's think step by step”) for the first 20 MMLU question sets. We were happy to see that the results match the other MMLU results from the paper, with chain-of-thought providing a moderate uplift on performance for both order dependent and order independent cases! This was our first run with chain of thought, with no tuning.

## Fig 2,  Impact of Enumeration

We re-ran the first 20 MMLU questions (our test set for Figure 4) with the numbers added. This shows improved performance across the board. We believe this is due to the models being trained with numbers on multiple choice questions.

## Fig 3, Long-Context Information Retrieval

Accuracy on extracting the correct pieces of information from a set of 10 documents. The 0 parallel batches case is the fully order dependent model, with others being different splitting locations for the parallel batches. Note that order dependent model shows the characteristic reduction in performance for documents further from the start. While the less order dependent inputs perform as well or better than the worst case for the order dependent inputs. The fully order independent inputs lead to significant degradation, as that is a full 10 parallel inputs. We hypothesize that this is due to the inputs being too far out of distribution. For this analysis we looked at Llama 3 8B Instruct as the non-instruct models struggled with this task.

## Fig 4, In-Context Learning

We implemented a sentiment classification task with in-context learning. The model was provided with 4 samples with labels, then asked to classify the fifth. The dataset is the Financial Phrase Bank [7] and with model attempting to classify them as positive or negative. To look at the effects of ordering on the model performance we always had three samples with the same label and 1 with the other label, with the different label always being first or last. When we ran this test, the in-context learning improved over the 1 shot test, showing that the models are using the examples. We did 4 batches of 100 questions, varying the valence of the 3 samples and the valence of the sample.

We found that putting the examples in parallel often improves the accuracy over the non-parallel case, or is as good. This was a very strong showing by our method and we thank the reviewer for suggesting it!

# Citations

See CaKN response for citations.

---

### Decision · Program_Chairs · 2024-09-25

**Decision:**

Accept (poster)

**Comment:**

This paper proposes a post-training method that makes an LLM robust to order variation in setups, such as multiple-choice questions, in which order should not affect its behaviour. The method is simple and theoretically justified.

The reviewers agreed that the problem is interesting and the perspective novel, but saw the method as more of an "engineering hack" whose general practical applicability is limited. They found moreover the method presentation and justification excessively complicated.

Still, after a lively discussion, all reviewers essentially agreed that the topic and approach are worth being disseminated.